# MoS_2_ Coexisting in 1T and 2H Phases Synthesized by Common Hydrothermal Method for Hydrogen Evolution Reaction

**DOI:** 10.3390/nano9060844

**Published:** 2019-06-02

**Authors:** Yixin Yao, Kelong Ao, Pengfei Lv, Qufu Wei

**Affiliations:** 1Key Laboratory of Eco-Textiles, Ministry of Education, Jiangnan University, Wuxi 214122, China; 18801510310@163.com (Y.Y.); aokelong_jnu@126.com (K.A.); 18861800875@163.com (P.L.); 2Fujian Key Laboratory of Novel Functional Textile Fibers and Materials, Minjiang University, Fuzhou 350108, China

**Keywords:** hydrogen evolution reaction, electrocatalysts, molybdenum disulfide

## Abstract

Molybdenum disulfide has been one of the most studied hydrogen evolution catalyst materials in recent years, but its disadvantages, such as poor conductivity, hinder its further development. Here, we employ the common hydrothermal method, followed by an additional solvothermal method to construct an uncommon molybdenum disulfide with two crystal forms of 1T and 2H to improve catalytic properties. The low overpotential (180 mV) and small Tafel slope (88 mV/dec) all indicated that molybdenum disulfide had favorable catalytic performance for hydrogen evolution. Further conjunctions revealed that the improvement of performance was probably related to the structural changes brought about by the 1T phase and the resulting sulfur vacancies, which could be used as a reference for the further application of MoS_2_.

## 1. Introduction

At present, our human energy deeply relies on fossil fuels. However, traditional energy is nonrenewable energy and causes serious pollution to the environment [1]. Owing to these issues, it is necessary to exploit and utilize renewable energy resources. Among numerous new types of energy, hydrogen, in the form of uncontaminated water, is considered a potential candidate as a combustion product with high gravimetric energy density [2,3,4]. Although Pt-group metals as electrocatalysts in the hydrogen evolution reaction (HER) is a most suitable method, searching for low-cost and abundant materials would be an essential key to realize the hydrogen energy age [5,6,7].

Lately, molybdenum disulfide (MoS_2_), one of transition metal dichalcogenides (TMDs) has had a broad appeal as an electrocatalyst due to its excellent performance for HER applications [8,9]. Naturally, MoS_2_ presents semiconducting properties in the 2H (hexagonal) phase, which has active sites only in the edges [10,11]. It transforms into the 1T (trigonal) phase with metallic properties and shows more outstanding HER performance, which was proven in previous works [12,13]. Presently, there are several methods to realize the phase transition, which is so-called phase engineering [14]. Thereinto, alkali metal intercalation is the mostly used technique [15]. In the process of reaction, an alkali metal, like lithium, would introduce electrons, causing the electron distribution to be changed among the original 2H phase. Then, the 2H phase would be converted into 1T phase for a more stable electron structure [16,17]. However, it may take quite a lot of time for the reaction and it could not be mass production. Electron beam irradiation has also been assumed in phase engineering by Lin and coworkers [18]. Moreover, a current study from Duerloo and coworkers [19] realized this conversion via mechanical deformation. Various techniques have been attempted to make MoS_2_ fulfill its structure transition, but it is still tough to utilize single 1T phase MoS_2_ in the consequence of the harsh reaction conditions or highly toxic reaction sources [20]. Therefore, provided that a special structure with the conductivity of 1T phase and the handleability of 2H phase is used, this problem will be solved. It has been revealed that the hybrid phases (1T@2H-MoS_2_) powerfully affect the electronic properties of the material in previous studies [13,21]. However, the impact of 1T@2H-MoS_2_ on the hydrogen evolution remains to be studied.

Herein, we adopted a facile strategy called two-step hydrothermal to dope the 1T phase into the 2H-MoS_2_. We observed that the fabricated MoS_2_, as an electrocatalyst, displayed excellent HER ability. After further exploration, sulfur vacancies, which could activate and optimize MoS_2_ basal planes for HER, were produced during the reaction [22]. Thus, the resulting MoS_2_ showed good hydrogen evolution catalysis through combination of the two aspects.

## 2. Materials and Methods

### 2.1. Synthesis of 1T@2H-MoS_2_

The 1T@2H MoS_2_ nanosheets were fabricated using a two-step hydrothermal method. Typically, 1 mmol (NH_4_)_6_Mo_7_O_24_·4H_2_O and 22 mmol thiourea were dissolved in 40 mL distilled water under vigorous stirring to form a homogeneous solution. After being stirred for 1 h, the solution was transferred into a 50 mL Teflon-lined stainless-steel autoclave and maintained at 200 °C for 20 h. Then, the reaction system was allowed to cool down to room temperature naturally. The obtained products were collected by centrifugation and washed with ethanol. Then, the stoichiometric MoS_2_ nanosheets were subjected to centrifugation and ultrasonication in ethanol solution and once more autoclavation under 220 °C for 8 h to form the 1T@2H-MoS_2_ and dried at 60 °C under vacuum.

### 2.2. Material Characterization

Scanning electron microscope (SEM) and corresponding Energy Dispersive Spectrometer (EDS) elemental mapping were taken with Nova NanoSEM NPE 207 (FEI, Hillsboro, OR, USA) on an accelerating voltage of 15 kV. X-ray photoelectron spectroscopy (XPS) was collected by an ESCALAB 250 Xi XPS (Thermo fisher scientific, Waltham, MA, USA) using Kα radiation for excitation source on a voltage of 2 V. Raman spectroscopy was collected by a Renjshaw inVin Raman microscope (Renishaw, London, UK) at a voltage of 15 kV. The X-ray diffraction (XRD) characterization was taken with RigakuUltima IV (Rigaku, Tokyo, Japan) through Cu Kα radiation.

### 2.3. Electrocatalytic HER Measurements

A three-electrode glass cell coupled up to a CHI 760 D (Shanghai Chenhua Instrument, Shanghai, China) was applied to all HER tests. Graphite rod and saturated calomel electrode (SEC) were chosen as counter electrode and reference electrode, respectively. The working electrode, a 5 mm diameter glassy carbon (GC) electrode daubed with a thin catalytic material, was exploited in our work. All potentials in the paper were converted to the reversible hydrogen electrodes (RHE): E_(RHE)_ = E_(SCE)_ + (0.242 + 0.059 pH) V. The process of making 1T@2H-MoS_2_ and 2H-MoS_2_ working electrodes was as follows. The catalyst of 4 mg was uniformly dispersed in the mixed solution of 10 mL ethanol and 20 μL Nafion solution by ultrasonication. After that, 25 μL of above solution was dropped onto the surface of the glassy carbon electrode by pipette and dried naturally. The ultimate catalyst loading was up to 0.875 mg/cm^2^.

## 3. Results and Discussion

### 3.1. Structural Analysis

The synthesis of the 1T phase doped into the 2H-MoS_2_ nanosheets (hereafter called 1T@2H-MoS_2_ nanosheets) involved a two-hydrothermal method (Scheme 1). The template 2H-MoS_2_ nanosheets were achieved by widely used hydrothermal process of (NH_4_)_6_Mo_7_O_24_·4H_2_O and thiourea in a deionized water at 200 °C. After centrifugation and ultrasonation, the MoS_2_ nanosheets were immersed in the ethanol solvent for the essential second solvothermal reaction at 220 °C. During this course, the sulfur vacancies was formed because of the generated hot temperature as the induction factor [23]. The existence of sulfur vacancy changed the density of the surrounding electron cloud, thus causing Mo to move and form a 1T configuration, and finally develop coexistence structure.

First of all, we used the scanning electron microscopy (SEM) to observe whether some changes in the structure would lead to changes in intuitive morphology. Figure 1a,b respectively show 2H molybdenum disulfide obtained by a hydrothermal synthesis. It can be seen that most of the lamellae of the layers are clustered together to form nanoflowers, while the morphology changes little after the secondary hydrothermal treatment. Therefore, other means are needed to verify the change in structure.

To further verify the hypothesis, X-ray photoelectronscopy (XPS) was used to analyze the diverse phase compositions in MoS_2_. Figure 2a shows the Mo 3d regions for the 2H-MoS_2_ which synthesized from the first hydrothermal step. It can be seen that there are two distinct peaks in the 232.6 eV and 229.4 eV, which correspond to the 3d_3/2_ and 3d_5/2_ of the Mo^4+^, thus indicating the presence of the 2H phase in molybdenum sulfide. In addition, the weak peak in 226.5 eV is S 2s of MoS_2_. In contrast with the one-step hydrothermal 2H-MoS_2_ nanoflowers, the 1T@2H-MoS_2_ nanosheets in the Mo 3d regions were also shown in Figure 2b. Through analyzing the Mo 3d spectrum of 2H-MoS_2_ nanoflowers, the high-resolution spectra are deconvoluted into seven peaks. Thereinto, a weak peak around 226.1 eV is S 2s of MoS_2_ and two main peaks at 232.6 eV (Mo 3d_3/2_) and 229.38 eV (Mo 3d_5/2_) are contributed to the oxidized Mo (IV) for 2H-MoS_2_. However, it is still not clear that the intensities of Mo 3d_3/2_ and Mo 3d_5/2_ overshoot changes in the opposite direction to the initial 2H-MoS_2_ synthesized from the first hydrothermal step. More importantly, it is noticed that the relatively weak doublet situated in binding energy at 231.7 eV and 228.1 eV are assigned to Mo^4+^ 3d_3/2_ and Mo^4+^ 3d_5/2_ of the 1T-MoS_2_, respectively. It indicates that the MoS_2_ exists the combination of 1T and 2H phase after the second solvothermal reaction. It is not hard to see that the 1T phase characteristic peak area is smaller than the 2H phase. Therefore, just a small percentage of 2H phase is converted into 1T phase. In addition, compared with the 2H-MoS_2_, the intensity of the S 2s shrunk a bit. This phenomenon may be caused by the sulfur vacancy produced in the second step reaction. Therefore, the presence of sulfur vacancy leads to the destruction of Mo-S, which causes Mo to combine with oxygen to form MoO_x_. That is why the peak intensity of MoO_x_ is much higher than that of 2H-MoS_2_. Of course, placing the sample in the air for a long period of time also increases the intensity. The certain 1T and 2H phase could be also confirmed from the X-ray diffraction (XRD) pattern (Figure 2c). Four evident diffraction peaks at 2θ = 13.8°, 32.8°, 39.5°, and 58.3°are expected for the (002), (100), (103), and (110) planes of typical 2H-MoS_2_ (JCPDS card No. 37-1492). Nevertheless, the characteristic peaks of the 1T phase were not easy to find, most likely as a consequence of the typical peaks of 1T-MoS_2_ being nearly same to that of 2H-MoS_2_.

For purpose of analyzing the phase and composition of the sample, Raman spectroscopy was also employed (Figure 3). It turns out that two major E${}_{2g}^{1}$ (379 cm^−1^) and A_1g_ (404 cm^−1^) activation modes can be clearly found for the 2H phase MoS_2_, and E${}_{2g}^{1}$ is due to the in-plane vibration of S and Mo atoms, A_1g_ is attributed to the relative vibration of S atoms in the out of plane direction [24,25,26]. From the point of view of peak intensity, A_1g_ mode is much stronger than E_1g_ mode, that is to say, it is more inclined to produce vertical plane (A_1g_), which may result in the edge-terminated structure [27]. The other two relatively weak peaks of 283 cm^−1^ and 454 cm^−1^ belong to E_1g_ and Longitudinal acoustic phonon mode. These four peaks correspond exactly to the 2H-MoS_2_. The spectrum drawn from the red line shows that on the basis of the original four peaks, there are two new very weak peaks at 219 cm^−1^ and 335 cm^−1^, respectively. These changes are basically derived from the changes in the structure of the material itself, and the two peaks are coincided with the 1T phase molybdenum disulfide. It indicates that during the second solvothermal reaction, the change of the temperature and solvent would impel the 2H phase structure to be unstable and transform into 1T phase to some extent for a better stability. Therefore, an optimal structure with coexisting 2H and 1T phases was formed in the end.

### 3.2. Catalytic Hydrogen Evolution

In order to examine the hydrogen evolution reaction ability, the synthesized 1T@2H-MoS_2_ nanosheets were loaded on glassy carbon disk as working electrodes in the typical three-electrode system with 0.5 M H_2_SO_4_. Saturated calomel electrode (SCE) was choose as reference electrode, so potentials were showed VS reversible hydrogen electrode (RHE). To avoid the effect of platinum dissolution on the whole reaction, the graphite rod was selected as counter electrode [28]. Meanwhile, for better comparison, commercial Pt/C and on-step 2H-MoS_2_ were also investigated in the same way. From the linear sweep voltammetry (LSV) in Figure 4a, the Pt/C electrode exhibits nearly 0 V onset potential, indicating that it indeed possesses excellent HER catalytic. According to one-step synthesized 2H-MoS_2_, it requires a large onset overpotential of 0.27 V. Contrary to 2H-MoS_2_, the improved synthesized 1T@2H-MoS_2_ possesses superior HER performance, which onset overpotential is decreased to 0.18 V. Consequently, the lower onset overpotential reflects the easier electrochemical reaction. The value of overpotential at 10 mA/cm^2^ is frequently used as an indicator of judging catalytic capacity. In the same way, the overpotential of 2H-MoS_2_ is not comparable to that of 1T@2H-MoS_2_. It shows that 1T phase intervention does make the catalytic performance elevate. The intrinsic properties and catalytic reaction efficiency of the catalyst could be demonstrated in the Tafel plots transformed from the polarization plots (Figure 4b). Importantly, liner potions are fitted to the Tafel equation (η = a + b log j, where b represents Tafel slop). The Tafel slop of Pt/C in our study (32 mV/dec) and previous studies are in accord [29,30]. The catalyst 1T@2H-MoS_2_ displays 88 mV/dec of Tafel slope, which is heavily dropped compared to 2H-MoS_2_ (136 mV/dec) and implies a favorable kinetic of 1T@2H-MoS_2_. As shown in Table 1, this catalyst is also comparable to other MoS_2_-based HER catalysts. Beyond that, the Tafel slope is related to the reaction mechanism. It could be also inferred that 1T@2H-MoS_2_ catalyst is in accordance with Volmer-Heyrovsky mechanism, and the rate-determining step is the desorption step (also called Heyrovsky) in consequence of the location between 40–120 mV/dec [30].

In order to calculate the electrochemical active area accurately, the double-layer capacitance between electrode and electrolyte was recorded through scanning cyclic voltammetry curve (Figure 5a,b). Then, using the current density and scanning rate at the intermediate potential value, a straight line is achieved. ∆j is the difference between anodic and cathodic current densities at 0.4 V vs. RHE (the dotted lines in the middle). Half of the slope of straight line is so-called double-layer capacitance (C_dl_), which reveals a much higher slope and lager active area of 1T@2H-MoS_2_ compared with 2H-MoS_2_ (Figure 5c).

### 3.3. Catalyst Durability

Durability is another prominent index to evaluate hydrogen evolution capability of electrocatalysts. Here, we adopt a method called chronoamperometry to present the current-time response curve (Figure 6), which shows that the catalyst has a realer activity for hydrogen precipitation and is most close to the practical application. This course was conducted in 0.5 M H_2_SO_4_ with potential of −0.6 V vs. SCE for 20 h. It is evident that current density attenuation is still extremely small. As expected, 1T@2H-MoS_2_ owns excellent durability.

### 3.4. HER Enhancement Mechanism

The novel 1T and 2H phase coexistence structure could boost HER ability, but it is necessary to understand the enhancement mechanism. As we know, ∆G_H_ could be used as an indicator for describing catalytic activity for various systems in hydrogen evolution reaction [38,39,40]. Just like the volcano plot [10], the most expected value of ∆G_H_ is about 0 eV. The higher hydrogen binding energy is not great enough to absorb protons tightly on the catalytic active sites. However, if it has weak binding energy, excess protons and catalytic will be closed as soon as possible and the desorption process is tough to conduct [11]. Therefore, the above two situations will all lower the rate of reaction. As for MoS_2_, it can be seen at the right side of the deviation point, that is to say, the main reason for the poor HER performance is its too large Gibbs free energy of hydrogen adsorption, so it is necessary to reduce the free energy through additional conditions.

According to previous studies, the introduction of sulfur vacancies could reduce the band gap of Mo atoms which play a major role in MoS_2_, namely the existence of sulfur vacancies promotes the generation of new band gaps [21,40]. The band gap of this vacancy is analogous to make the valence band closer to the Fermi energy level, which reflects the effect of *n*-type doping. Further, the narrower the band gap, the lower the energy required for the valence electron transition, which is more profitable to the creation of hydrogen adsorption environment in the place where the vacancy is generated, and ultimately reduces the free energy. In addition, according to the Table 2, the analysis of XPS in the second step hydrothermal reaction also proved the formation of sulfur vacancy. This is consistent with the previously reported temperature [22], which can induce the transformation of part of 2H to 1T in molybdenum disulfide, although so far it has not been possible to accurately explain the formation mechanism. However, it is supposed to be related to the increase of valence electron energy and the activation of sulfur atoms due to the rise of temperature. In addition, changes in solvents may also cause phase transition. we synthesized the sample in water instead of ethanal in the second solvothermal reaction. After finishing the reaction, we opened the Teflon lining and could clearly observe that the molybdenum disulfide powder remained deposited at the bottom and that the water was still clear. In the previous reaction, the solution appeared as a turbid liquid with black particles dispersed in the solution, and black powder adhered to the side of the lining. The XPS analysis also proved that the 1T phase did not form in the second hydrothermal reaction (Figure 7). Therefore, both the temperature and the solvent affect the phase transition.

On the other hand, the deficiency of 2H phase MoS_2_ in application is gradually discovered. It is found that the performance of MoS_2_ would be greatly improved after the transformation of 2H phase to 1T phase induced by the ambient environment. The diversities between the two phases are mainly reflected in the following aspects. First, properties of conductor or semiconductor would have appeared if the MoS_2_ is 1T or 2H phase [41,42,43,44]. Improved conductivity not only means faster transmission of electrons, but also a condition favorable to the reaction. Second, active sites of initial 2H-MoS_2_ are just confined to marginal position [31,37]. However, converting into 1T-MoS_2_, basal plane would be excited to a wider range of active sites [7,45]. Finally, in the HER, protons in the solution first adhere to S atoms in 2H-MoS_2_ to form S-H, and then shift to Mo atoms to product Mo-H. 1T-MoS_2_ has a more direct path to generate H_2_, in more detail, it can straightly absorb protons on Mo atoms [41,46]. These differences impel 1T phase superior to 2H phase when used as catalyst. For the 1T@2H-MoS_2,_ although we could not explain the HER mechanism, we may suspect that protons bind not only directly to the Mo sites, but also to the S sites.

## 4. Conclusions

In a nutshell, a unique structure of molybdenum disulfide was synthesized in a simple hydrothermal method and used as a highly efficient catalyst in hydrogen evolution reaction. The sulfur vacancy formed in the secondary solvothermal not only shortened the forbidden band width of molybdenum disulfide, but also promoted the transition from 2H to 1T phase. The overall catalytic performance of MoS_2_ was improved by the changes of these two key structures. As a matter of fact, the compound possessed outstanding HER ability with low overpotential and large current density. Furthermore, other similar chemical compound could also draw on this technique for opening new opportunities in different areas.

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
