# Peer review of "MoS2Â Coexisting in 1T and 2H Phases Synthesized by Common Hydrothermal Method for Hydrogen Evolution Reaction"

_nanomaterials, 2019, doi:10.3390/nano9060844_

Round 1
Reviewer 1 Report
I am satisfied with the revision and now recommend its acceptance.
Reviewer 2 Report
The authors have nicely addressed my queries. I can therefore recommend it for publication.
This manuscript is a resubmission of an earlier submission. The following is a list of the peer review reports and author responses from that submission.
Round 1
Reviewer 1 Report
This manuscript demonstrates MoS2 coexisting in 1T and 2H phases synthesized by common hydrothermal method for hydrogen evolution reaction in acidic condition. I think the manuscript id premature to be published in Nanomaterials. Specific comments are following;
Is the amount of the prepared catalyst in the ink 4 mL, not mg?
Is the centrifugation earlier than ultrasonication step?
In the scheme 1, the temperature of the second hydrothermal step does not match with the text.
In XPS, the authors mentioned that all peaks in Mo 3d regions are blue shifted. What is the standard of the shift? In addition, I recommend that the authors give the XPS analysis for the 2H MoS2 which can obtain from the first hydrothermal step and then, compared the result with 1T@2H-MoS2.
I think the authors focused on the improvement of stability rather than the HER activity because the activity is too low. 1T MoS2 is metallic but unstable, while 2H MoS2 has low activity but stable. Thus, they made the mixed phase of them. However, I don’t know the stability is really improved or not. Therefore, comparison the stability of 1T@2H MoS2 with only 1T MoS2 should be provided.
The current density in Figure 6 is too low because it was measured at – 0.7 V vs. SCE then, the current density should match the current density in LSV curves. (at least more than 10 mA cm-2)
Is the synthesis reproducible? Moreover, what is ratio between 1T MoS2 and 2H MoS2 and is the content of 1T MoS2 controllable? Is the content of 1T MoS2 optimized?
The authors claim that the conversion of the phase from 2H MoS2 to 1T MoS2 in the second hydrothermal step is due to the rise of temperature. However, I don’t agree with it because the solvent is different from the first step. If you want to claim like that, please synthesize the sample in water instead of ethanol.
Please, cite the recently published papers.
I think Figure 7 cannot be in main data because there are no data from your experiments. You can cite the reference 10 as you wrote.
Is there any evidence that the sulfur vacancy is formed in the secondary solvothermal step?
Reviewer 2 Report
Main criticism: It does not appear to me that the MoS2 catalyst made is good for HER at all. One would need to apply an overpotential of 700 mV to obtain currents of 4-5 mA/cm2 (Figure 6). I would like to suggest that the authors discuss how HER occurs mechanistically on the MoS2. Is it through the Mo or S sites?